# The association between maternal body mass index and serial plasma oxytocin levels during labor

Anna Ramö Isgren[1], Sara Carlhäll[1], Mark Dennis Retrato[2], Chamali Kodikara[2], Kumari A. Ubhayasekera[2], Preben Kjölhede[1], Jonas Bergquist[2], Marie Blomberg[1]*

1 Department of Obstetrics and Gynecology and Department of Biomedical and Clinical Sciences, Linköping University, Linköping, Sweden, 2 Department of Chemistry-BMC, Analytical Chemistry and Neurochemistry, Uppsala University, Uppsala, Sweden

☯ These authors contributed equally to this work.

* marie.blomberg@liu.se

**Data Availability Statement:** All relevant data for this study are publicly available from the Dryad repository (https://doi.org/10.5061/dryad.kprr4xh77).

## Abstract

### Objective

To evaluate the association between maternal body mass index (BMI) and plasma oxytocin (OT) levels at different OT infusion rates in labor.

### Methods

A prospective observational study analyzing serial plasma samples in laboring women with OT infusion. The women were categorized into three groups, women with non-obesity (BMI 18.5–29.9, n = 12), obesity (BMI 30.0–34.9, n = 13), and morbid obesity (BMI $\geq$ 35.0, n = 15). Plasma OT was analyzed using tandem mass spectrometry.

### Results

Except for a low positive correlation between OT levels and BMI and significantly increased plasma OT levels in women with morbid obesity at the OT infusion rate of 3.3 mU/min, no significant differences in OT levels between the BMI groups were found. Further, the inter-individual differences in OT levels were large and no dose-dependent increase of OT levels was seen.

### Conclusions

Other factors than plasma OT levels may be more likely to determine the clinical response of OT infusion in women with obesity. Perhaps the observed clinical need and individual response would be a better predictor of plasma OT levels than a pre-determined OT infusion rate. The OT dosage guidelines for labor augmentation should be individualized according to clinical response rather than generalized.

### Trial registration

**Clinical trial registration:** ClinicalTrials.gov ID NCT04093479.

**Funding:** Medical Research Council of Southeast Sweden (FORSS-909171 (SC) FORSS-931749 (MB)), ALF-grants, Region Östergötland (RÖ-966229 (ARI), RÖ-918761 (SC), RÖ-939997 (MB)), The Swedish Research Council (2015-4870 (JB)), The Erasmus+ Programme of the European Union (EACH EM JMD, project No 586571). The funders had no role in study design, data collection and analysis, decision to publish, or preparation of the manuscript.

**Competing interests:** The authors have declared that no competing interests exist.

## Introduction

The global epidemic of obesity, leading to increasing number of women in the reproductive age with increasing body mass index (BMI), is concerning given the obesity-related adverse consequences during pregnancy and labor. During labor, women with obesity are at an increased risk of complications indicating an impaired uterine contractility, such as labor arrest, emergency cesarean section and major postpartum bleeding [1–4]. The peptide hormone oxytocin (OT) is a key hormone during labor by promoting contractility of the uterine myometrium via specific OT receptors [5]. Because of its positive effect on uterine contractility, exogenously administered OT is used in obstetric care for induction of labor as well as for treatment of labor arrest and postpartum hemorrhage. Observational studies have shown that women with overweight or obesity require higher cumulative OT doses and higher OT infusion rates, compared with normal weight women, for normal labor progression [6–9]. It has been suggested that obesity-associated factors cause an inhibitory effect on myometrial contractility with a reduced response to OT [10,11]. Furthermore, it has been suggested that recommended doses of exogenously administered OT for labor augmentation, is insufficient in women with obesity, due to an increased volume of distribution [7]. However, few studies have analyzed OT levels in pregnant women in labor with OT infusion and only one has related the results to maternal BMI [12,13]. These studies present varying results, and all have used immunoassay methods) for the OT analyses. In contrast to immunoassay methods, tandem mass spectrometry (MS/MS) offers a highly specific and sensitive detection method for absolute quantitation of OT with low risk of interference [14,15]. To our knowledge, there are no previous studies with serial measurements of maternal OT levels during labor analyzed with MS/MS in relation to maternal BMI. Therefore, in this study we aimed to evaluate the association between maternal BMI and plasma OT levels at different infusion rates of OT in the first stage of labor, analyzed by MS/MS. We hypothesized, that women with obesity, requiring OT infusion in labor, would have lower plasma OT levels than women with normal weight or overweight during equal doses of OT infusion.

## Materials and methods

### Study design and study participants

The Oxytocin in Labor In Vivo (OLIV) trial, was a prospective observational pilot study, conducted at Linköping University Hospital, Sweden. Inclusion criteria were $\geq$ 18 years old, singleton pregnancy, $\geq$ 37 gestational weeks, cephalic presentation, and indication for OT infusion during the first stage of labor. Indication for treatment with OT infusion followed the recommendations in the current Swedish guidelines [16].

The women were categorized according to BMI, based on weight on admission to the labor ward, in three groups: women with non-obesity (BMI 18.5–29.9 kg/m2), with obesity (BMI 30.0–34.9 kg/m2), and with morbid obesity (BMI $\geq$ 35.0 kg/m2). The recruitment of study participants, blood sampling and plasma separation was carried out around the clock by a research team of specially trained midwives and obstetricians at the labor ward during three weeks between October 2019 and January 2020.

The women who fulfilled the inclusion criteria received oral and written information about the purpose and concept of the study. Written informed consent was obtained from the study participants. Data on maternal and obstetric characteristics were extracted from the woman´s medical records, and included age, parity, smoking in first trimester, diabetes mellitus, hypertension disorder, BMI on admission to the labor ward, gestational age at delivery, use of epidural analgesia, mode of delivery and fetal birth weight.

The women were included in the study when treatment with OT for labor augmentation was indicated. After inclusion, before the OT infusion was started, a peripheral venous catheter was placed in the contralateral arm from where OT infusion was administered. The OT infusion was prepared to a concentration of 10 mU/mL (16.67 ng/mL) and was given according to a standardized protocol with an infusion rate starting at 3.33 mU/min with an increase in rate of 3.33 mU/min every 20 minutes.

The progress of labor and the number of uterine contractions (maximum 5 per 10 minutes) determined the infusion-rate. Blood samples for measuring plasma OT were taken prior to the initiation of OT infusion and 20 minutes after each subsequent increase of the OT infusion. Repeated sampling continued until the infusion rate was not further increased, was decreased, or the infusion was turned off, or when labor reached the second stage (cervix fully dilated).

## Blood sampling

Venous blood samples were collected without stasis in 6 mL BD vacutainer® K2-EDTA tubes from the peripheral venous catheter in the contralateral arm from where OT infusion was administered. These were inverted gently, immediately placed on ice, and centrifuged at 2000 x g for 10 minutes. The plasma was separated into 1 mL plastic tubes, coded and stored at minus 70 ˚C. The OT plasma analysis was performed on the deidentified and coded samples at the Department of Chemistry–Biomedical Centre, Uppsala University, Uppsala, Sweden.

## Extraction of OT

A 200 μL of human plasma sample was mixed with 20 μL of 100 ng/mL OT-$d_5$ (Merck, Sweden), which served as the internal standard. All other chemicals and solvents were of MS and analytical grade (Merck, Sweden). 2M ammonium acetate 100 μL and 200 μL ice cold acetonitrile were added and the resulting solution was vortexed for 10 minutes. The solution was centrifuged for 15 minutes at 20,000 ×$g$ at 4 ˚C. The supernatant was collected and dried under a gentle stream of nitrogen. The dried supernatant was reconstituted with a 75:25 (v/v) acetonitrile-water mixture, prior to analysis using ultra performance liquid chromatography (UPLC)-Orbitrap MS/MS.

## UPLC-Orbitrap MS/MS analysis of OT

Chromatographic separation of the targeted OT was achieved by using an Acquity UPLC system (Waters, Milford, USA), equipped with a Kinetex column (2.6 μm, 100 mm × 2.10 mm) (Phenomenex, Torrance, CA, USA). The elution gradient was carried out with a binary solvent system consisting of ultra-pure water (solvent A), and acetonitrile (solvent B). Both mobile phases contained 0.1% acetic acid. The applied linear gradient profile started with 85.0% A and 15.0% B, for 0.50 min. Then, in 1.50 min, the proportion of solvent B was increased to 30.0%. This mobile phase condition was kept for 3.0 min and was changed to 15.0% B at 3.50 min. At this point, the mobile phase was switched back to the initial conditions and the column was allowed to re-equilibrate for 7.0 min. The flow rate was kept constant at 0.30 mL min−1. The oven temperature was set at 40˚C. A 10 μL aliquot of each sample was injected for analysis.

OT identification was achieved on an Orbitrap mass spectrometer (Thermo Fisher Scientific, San Jose, USA), equipped with a heated electrospray ionization probe, operating in positive ionization mode with single ion monitoring. Ionization source working parameters were optimized and involved the sheath gas flow rate (40 au, arbitrary units), auxiliary gas flow rate (20 au), sweep gas flow rate (5 au), spray voltage (3.5 kV), spray current (5.30 μA), capillary temperature (350˚C), and auxiliary gas heater temperature (350˚C). The method's analytical

time was seven minutes, and it was implemented in single ion monitoring mode. The selected OT parent ion was at m/z 1007.4454 and the monoisotopic ion was at m/z 504.2261 (OT + 2H +)2+. The corresponding internal standard OT-d5 parent ion was at m/z 1012.4751 and its monoisotopic ion was at m/z 506.2264 (OT-d5 + 2H+)2+. The monoisotopic ions for OT and OT-d5 represent the entire OT molecule that is doubly charged. The resolution employed was at 35,000, while the automatic gain control target was set at the high dynamic range (5×104 ions), and the maximum injection time was 200 ms. The option of "all ion fragmentation" using the high energy collision dissociation cell was only used to investigate the confirmation potential of generated fragments and was turned off during the actual analysis. The method was optimized and validated according to the European Medicine Agency ICH guidelines [17]. OT and its deuterated internal standard were injected directly into the mass spectrometer and were adjusted for molecular fragmentation. This was done using 2 g/mL methanol and solution to determine molecular specific ionization in either positive or negative electrospray ionization modes. All fragmentations were examined for cross talk as well as specificity. Finally, a particular multiple reaction monitoring was chosen. To maximize the signal and S/N ratio, the ion source and spectrometric parameters were manually tuned. Electrospray ionization modes, on the other hand, were chosen as an appropriate ionization method for the analysis of OT.

The linearity of the OT was evaluated over a range of concentrations (0.1–10000 ng/mL) and the correlation coefficient (r2) was 0.9993. The limit of quantification (LOQ, signal-to-noise ratio = 10) and coefficient of variation of OT were 0.05 ng/mL, and less than 15%, respectively. The precision of the OT quantitation was validated by running quality control samples in six replicates on the same day and on three independent days. The intra-assay CV was less than 10%, while the inter-assay—coefficient of variation was less than 15%. The recovery of the OT in the assay was determined to be more than 85%. Initial instrument calibration was achieved by infusing calibration mixtures (Thermo Fisher Scientific, San Jose, USA) for the positive and negative ion modes. Instrument control and data processing were carried out by Xcalibur Processing Setup Quan software (Thermo Fisher Scientific,San Jose, USA). A total number of 207 blood samples were taken and of these, 205 samples could successfully be analyzed with detectable levels of OT, the LOQ was 0.05 ng/mL, and the relative standard deviation was less than 20%.

## Statistical analyses

The statistical analysis was performed using the software IBM SPSS version 26 (IBM Corporation, Armonk, NY, USA). Categorical data are presented as number and percent. Continuous data are presented as mean and one standard deviation (SD), or median and inter quartile range (IQR) if not normally distributed. Wilcoxon signed-rank test was used to compare related medians at different sampling occasions. Categorical maternal and obstetric characteristics were compared over the BMI strata using a Person´s Chi$^2$ test or Fischer´s exact test, as appropriate. One-way analysis of variance (ANOVA) was used for comparison of normally distributed continuous variables and Kruskal-Wallis ANOVA for not normally distributed continuous variables. Bivariate correlations were evaluated using the Spearman´s Correlation coefficient. The significance level was set at 0.05 for two-tailed test.

## Ethical approval

The study received ethical approval on June 5[th], 2019, Dnr 2019–03007 by the Swedish Ethical Review Authority.

## Results

The study population included 40 women; 12 women (30%) were categorized with non-obesity, 13 women (32.5%) with obesity, and 15 women (37.5%) with morbid obesity (Table 1).

Fig 1 accounts for the number of women in each BMI category, available for blood sampling before the OT infusion started, and at the different OT infusion rates from 3.3 mU/min until 10.0 mU/min. The median number of serial samples per study participant was 4 (range 1–16 samples) (S1 Table).

The correlation between BMI and plasma OT levels was examined before and after each increase of the OT infusion rate (Fig 2A–2D). A significant correlation was found between BMI and plasma OT at the OT infusion rate of 3.3 mU/min (Spearman´s correlation coefficient 0.333, p = 0.036). At higher rates of OT infusion, no significant correlation between BMI and plasma OT was seen.

Fig 3A–3D demonstrate the median, interquartile and the range of plasma OT levels before and at each increase of the OT infusion rate until 10.0 mU/min. At an OT infusion rate of 3.3 mU/min there was a significantly higher median OT level in the women with obesity (median 16.04 ng/mL (IQR 9.78; 28.82 ng/mL) and morbid obesity (median 24.61 ng/mL (IQR 10.69; 42.42 ng/mL) compared with the women with non-obesity (median 6.97 ng/mL (IQR 5.23;

**Table 1. Maternal and obstetric characteristics according to categorization of BMI.**

| Characteristics | Non-obese BMI 18.5–29.9 (n = 12) | Obese BMI 30.0–34.9 (n = 13) | Morbidly obese BMI ≥ 35.0 (n = 15) | p |
|---|---|---|---|---|
| Maternal age (years) | 30.3 ± 2.4 | 30.0 ± 5.1 | 32.5 ± 6.2 | 0.369 |
| Smoking in first trimester | 0 (0.0) | 0 (0.0) | 0 (0.0) | NA |
| Diabetes mellitus | 1 (8.3) | 0 (0.0) | 0 (0.0) | 0.302 |
| Hypertension (including preeclampsia) | 0 (0.0) | 1(7.7) | 4 (26.7) | 0.093 |
| Maternal height (cm) | 164.9 ± 7.6 | 165.6 ± 8.4 | 163.7 ± 7.5 | 0.814 |
| Weight at delivery (kg) | 72.2 ± 7.6 | 86.8 ± 9.2 | 106.3 ± 16.4 | <0.01 |
| BMI on admission (kg/m2) | 26.5 ± 2.1 | 31.5 ± 1.1 | 39.5 ± 4.1 | <0.01 |
| Parity | | | | |
| Nulliparous | 7 (58.3) | 8 (61.5) | 7 (46.7) | 0.705 |
| Previous cesarean section | 1(8.3) | 3 (23.1) | 3 (20.0) | 0.594 |
| Gestational age (days) | 280 ±10 | 282 ±9 | 283 ±9 | 0.679 |
| Epidural anesthesia | 11 (91.7) | 12 (92.3) | 14 (93.3) | 0.986 |
| Mode of delivery | | | | 0.023 |
| Vaginal, non-instrumental | 11 (91.7) | 13 (100.0) | 8 (53.3) | |
| Vaginal, instrumental | 0 (0.0) | 0 (0.0) | 3 (20.0) | |
| Cesarean section | 1 (8.3) | 0 (0.0) | 4 (26.7) | |
| Induction of labor (yes) | 5 (41.7) | 2 (15.4) | 10 (66.7) | 0.024 |
| Method for induction of labor | | | | 0.071 |
| Oral misoprostol | 2 (40.0) | 1 (50.0) | 7 (70.0) | |
| Balloon catheter | 3 (60.0) | 1 (50.0) | 0 (0.0) | |
| Amniotomy | 0 (0.0) | 0 (0.0) | 3 (30.0 | |
| Cervical dilatation at start of oxytocin | | | | |
| cm | 7 [4.25–8.0] | 7 [3.5–8.0] | 4 [4.0–6.0] | 0.071 |
| < 6 cm | 4 (33.3) | 5 (38.5) | 11 (73.3) | 0.071 |
| ≥ 6–10 cm | 8 (66.7) | 8 (61.5) | 4 (26.7) | |

Data are expressed as mean ± one standard deviation, median and [interquartile range], or number and (%). BMI, body mass index.

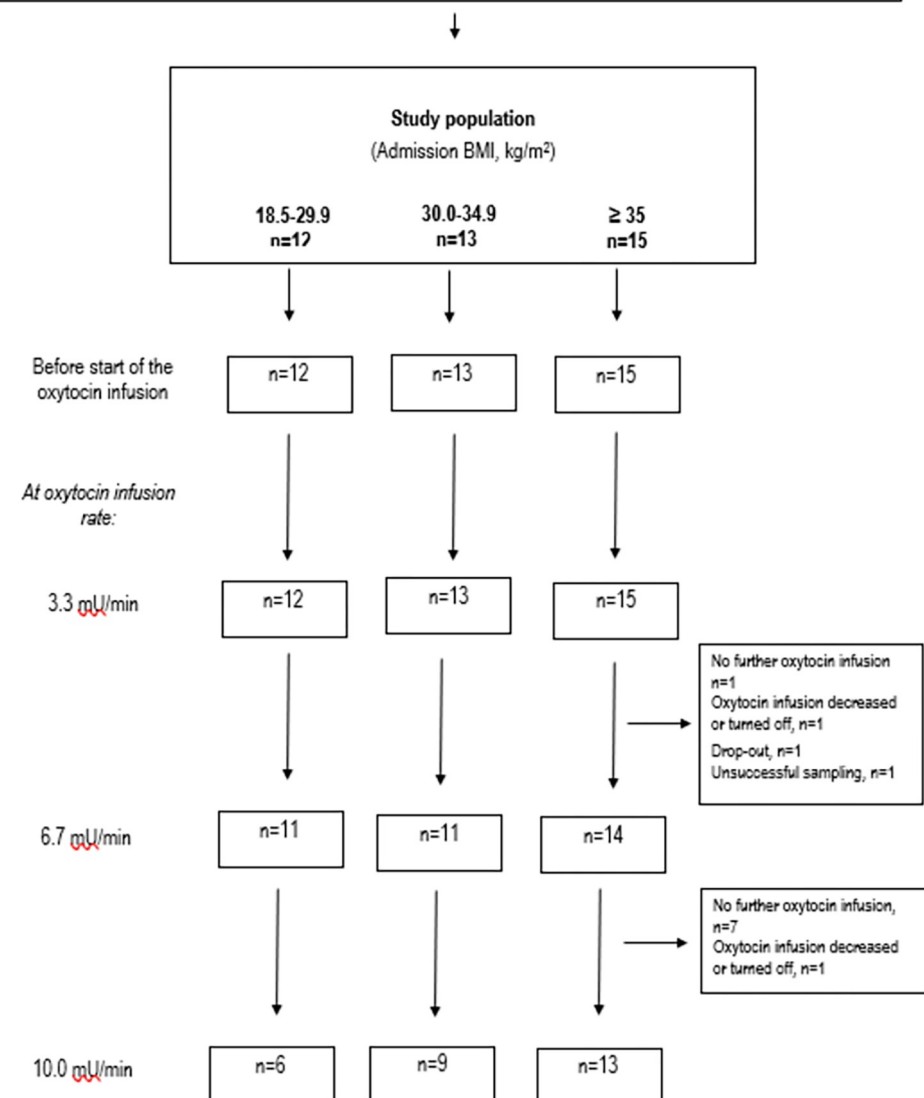

**Fig 1. Flowchart of the study population and sample collection.** Number of women in each body mass index (BMI) category available for blood sampling before the OT infusion started, and at the different OT infusion rates from 3.3 mU/min until 10.0 mU/min.

14.83 ng/mL) p = 0.013). According to the post-hoc test, this was mainly attributed to a significant difference between the women with non-obesity and the women with morbid obesity (p<0.05). Further, no significant differences in the median OT levels were found between the BMI groups before the OT infusion started at OT infusion rate 6.7 mU/min and 10.0 mU/min.

The differences in the pre-infusion and the post-infusion OT levels for each woman in the three BMI groups are presented in Figs 4A–4C and 5A–5C. The pattern in all BMI groups showed that the OT levels both increased and decreased after the initiation of OT infusion.

Fig 4A–4C shows the pre infusion OT level and post OT infusion level at 3.3 mU/min for each woman, in the three BMI groups. The higher the BMI category, the more women had

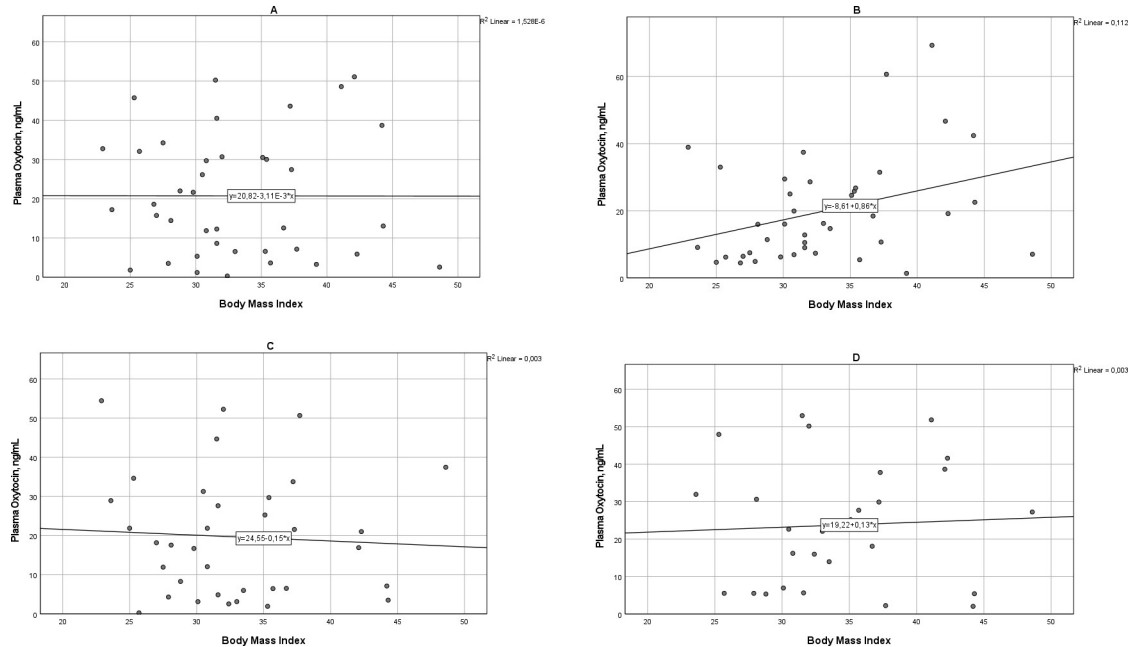

**Fig 2.** (A-D) Scatterplots presenting the correlation between maternal body mass index and plasma oxytocin. (A) Before start of oxytocin infusion (r = 0.048). (B) At oxytocin infusion rate 3.3 mU/min (r = 0.333). (C) At oxytocin infusion rate 6.7 mU/min (r = 0.049). D. At oxytocin infusion rate 10.0 mU/min (r = 0.017).

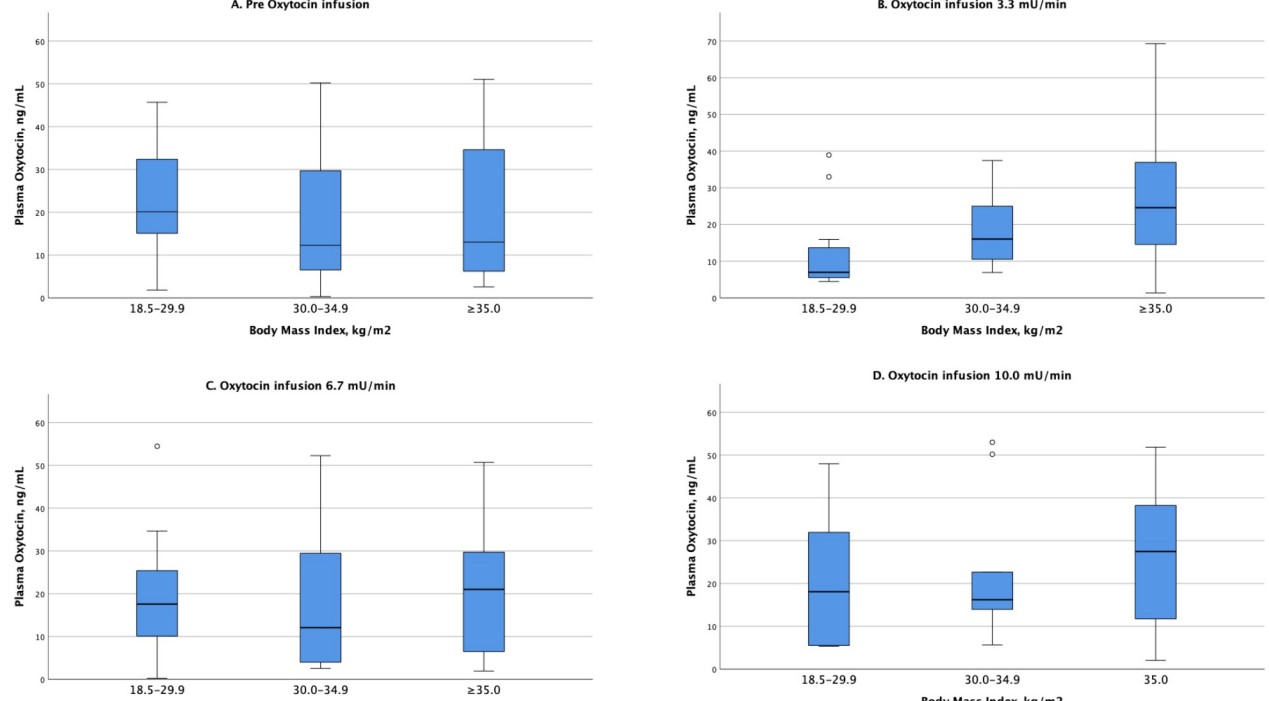

**Fig 3.** (A-D) Box-plots showing median plasma oxytocin levels in laboring women according to body mass index. The boxes indicate 25% and 75% percentiles and error bars maximum and minimum levels. Outliers are plotted as individual points. (A) Before start of oxytocin infusion (p = 0.788). (B) At oxytocin infusion rate 3.3 mU/min (p = 0.013). (C) At oxytocin infusion rate 6.7 mU/min (p = 0.891). (D) At oxytocin infusion rate 10.0 mU/min (p = 0.130).

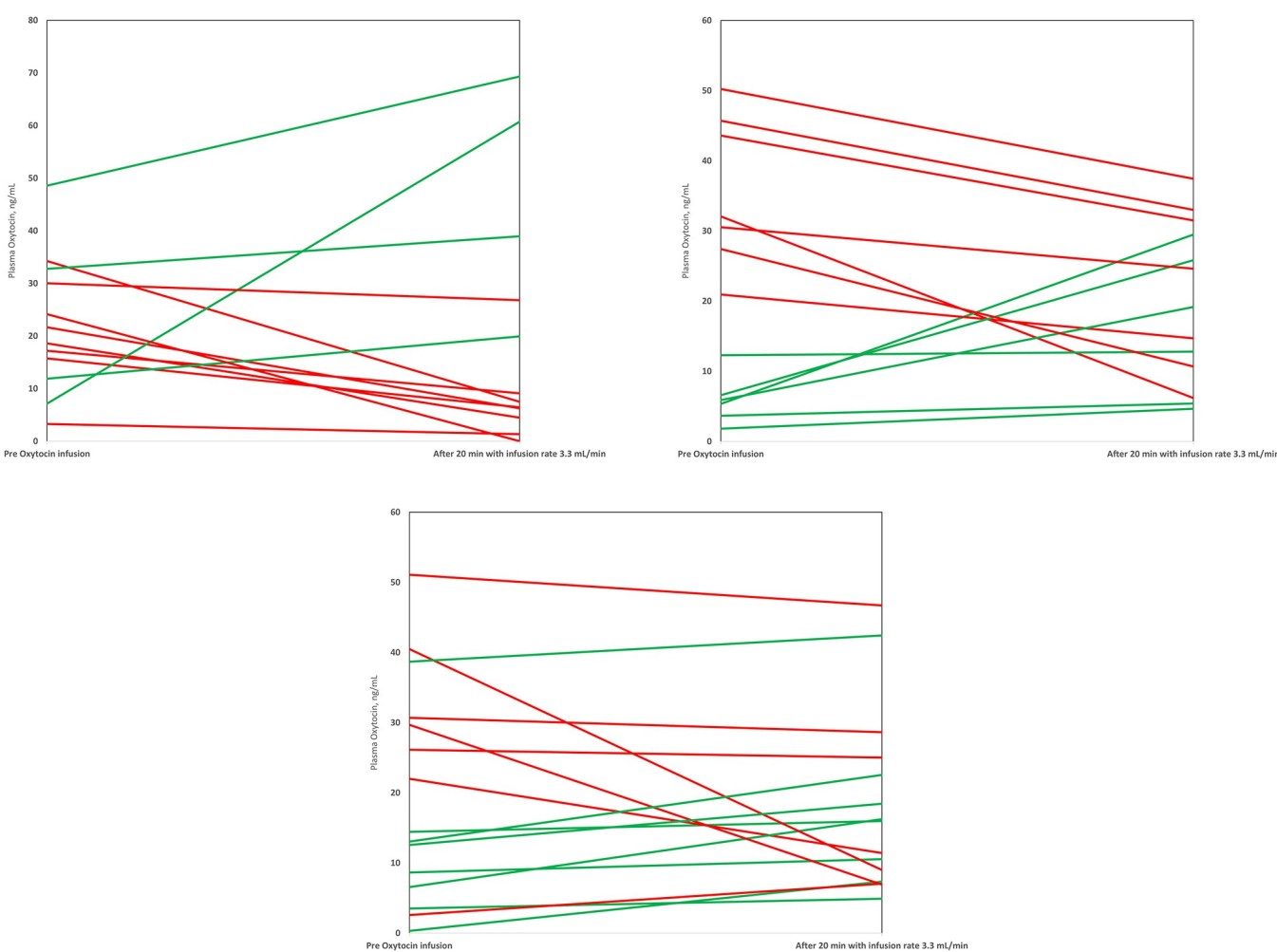

**Fig 4.** (A-C) Pre and post infusion plasma oxytocin level (ng/mL) for each woman in respective BMI category. The lines illustrate plasma oxytocin levels that rise (green colour) or lowers (red colour) from the pre oxytocin infusion level to the post infusion level, measured after 20 minutes at an oxytocin infusion rate of 3.3 mU/min in women with non-obesity (A), women with obesity (B) and women with morbid obesity (C).

increased post infusion OT levels at this infusion rate. In 4/12 (33%) of the women with non-obesity, the plasma OT levels increased compared with 6/13 (46%) of the women with obesity and 8/15 (53%) of the women with morbid obesity.

Fig 5A–5C shows the pre infusion OT level and the last OT measurement for each woman, in the three BMI groups. In the group of women with morbid obesity the OT levels increased from pre-OT infusion to the last measurement in 10/15 women (67%) compared with 6/12 (50%) women in the group with non-obesity.

Women with induction of labor differed between the BMI groups, with the highest rate among women with morbid obesity (Table 1). According to the post-hoc test, this was mainly attributed to a significant difference between women with obesity and morbid obesity (p<0.05). Median OT levels before OT infusion was started and at an OT infusion rate of 3.3 mU/min and 6.7 mU/min were compared between women with spontaneous onset and induced labor. No significant differences in median OT levels appeared at any time point, (p = 0.464, p = 0.871, p = 0.364, respectively). The cervical dilatation at start of OT infusion differed significantly between the BMI groups, however, no correlation was found between

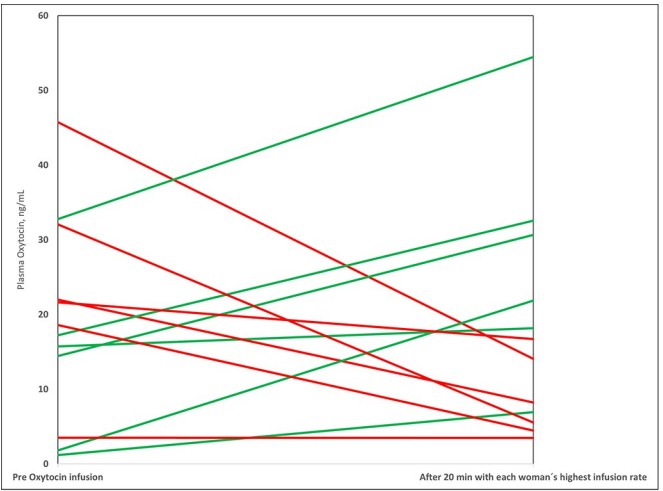
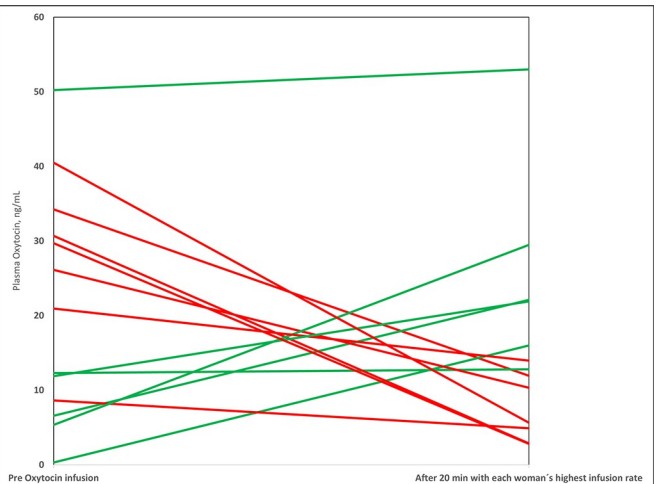
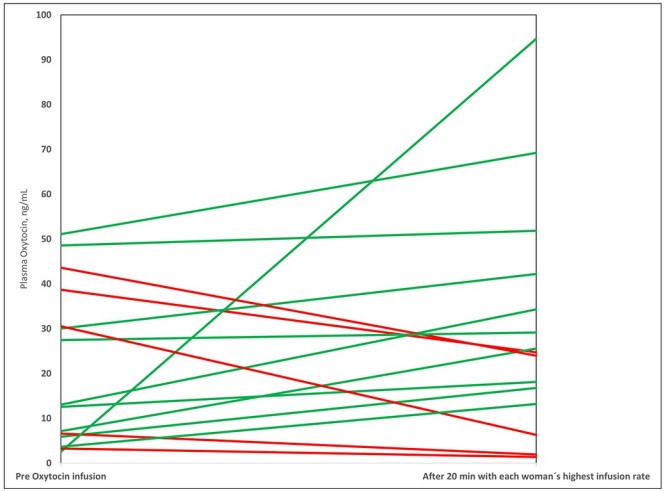

**Fig 5.** (A-C) Pre and last infusion plasma oxytocin levels (ng/mL) for each woman in respective BMI category. The lines illustrate plasma oxytocin levels that rise (green colour) or lowers (red colour) from the pre oxytocin infusion level to to the last measured oxytocin level in women with non-obesity (A), women with obesity (B) and women with morbid obesity (C).

cervical dilatation at start of OT infusion and median OT levels at any sampling point (Spearman´s correlation coefficient ranging from -0.002 to 0.124).

The OT levels showed high inter- and intra-individual variability in all BMI groups and no dose-dependent increase of OT levels in response to an increased rate of OT infusion was seen (S1 Table).

## Discussion

This prospective observational pilot study included 40 women in labor requiring OT infusion to enhance uterine contractions. Serial blood samples were taken in the first stage of active labor to evaluate the association between BMI and plasma OT levels.

Since women with obesity require higher OT infusion rates and doses [6–9], we hypothesized, that plasma OT levels in women with obesity will be lower compared with women with non-obesity during equal doses of OT infusion. Interestingly, except for a low positive correlation between OT levels and BMI and significantly increased plasma OT levels in women with

morbid obesity, at the OT infusion rate of 3.3 mU/min, no other significant differences in OT levels between the BMI groups were found. Further, the inter-individual differences in the OT levels were large, with both rising and falling post infusion OT levels. However, it seemed that with higher BMI category, the more women had rising post infusion OT levels, but the groups were relatively small, and interpretations must be done with caution.

Women with obesity have increased risk for labor arrest, emergency cesarean section, and major postpartum bleeding [1–4]. These adverse outcomes indicate that obesity has an inhibitory effect on uterine contractility. The results from the present study do not support the hypothesis that women with obesity have lower plasma levels of OT before or after OT infusion during labor. Hence, our data suggest that the reason for impaired uterine contractility in women with obesity are unlikely due to differences in plasma OT levels during OT infusion. Other explanations proposed may be that women with obesity have different metabolic profile, altered OT receptor signaling or myometrial OT receptor expression [10,18–21].

If the binding of OT to the myometrial OT receptor is impaired in women with obesity, the amount of circulating OT in the blood may be of minor importance to strengthen the uterine contractility in labor. Carlson et al. studied women with obesity and OT augmentation and found that women with obesity class II and III (morbid obesity) received increased mean OT infusion rates without difference in labor duration compared with women with obesity class I [22]. Thus, since women with morbid obesity did not have shorter labor duration despite an increased OT exposure, they might be less able to assimilate OT, resulting in less effective treatment of dysfunctional labor.

Furthermore, it is possible that obesity-related concentrations of adipokines might affect the OT—OT receptor complex. Leptin, an adipokine found in higher concentrations in pregnant women with obesity, has a cumulative inhibitory effect on myometrial contractility in vitro [23,24]. However, the mechanism by which leptin mediates this inhibitory effect on myometrial contractility is unknown.

No earlier study has investigated serial sampling of plasma OT during OT infusion related to the woman´s BMI. De Tina et al. analyzed paired samples of OT levels using the ELISA technique, in 12 women with obesity and 18 normal weight women, before the OT infusion was started and 20 minutes after the expected maximum rate of the OT infusion. In line with our results, no differences in the OT levels between the BMI groups were seen, and the OT levels showed high inter-individual variability, both at baseline and at the maximum OT infusion rate [12,15].

The theory that the plasma OT concentration has a natural fluctuation and large individual variability in labor, which has been shown in other studies analyzing maternal plasma OT in pregnancy and labor using immunoassay methods [12,25,26], seemed to be supported by our findings. The present study demonstrated more pronounced variations and higher levels of OT compared with the OT levels presented in earlier studies [12,13]. In most women, when the pre-OT infusion level was high, the OT level decreased after initiation of OT infusion and, on the contrary, when the pre-OT infusion level was low, the OT levels during OT infusion increased. This pattern was most pronounced among women with obesity. Prevost et al. measured maternal plasma OT levels in 272 women in early pregnancy, late pregnancy and postpartum. Similar to our finding, they found large within-subject variations of OT levels with rising and falling OT patterns between the measurements [25].

Another result in the present study, in line with the study by de Tina et al [12] was the absence of a dose-dependent increase of OT plasma levels in response to an increased rate of OT infusion regardless of BMI. The underlying mechanism for this finding is unknown and several factors may be of importance.

Since the measured substance OT is a composition of the endogenous OT secretion and the exogenously administrated OT, it is not possible to distinguish the amount of endogenous OT in plasma from the exogenously administrated OT in the analysis. The endogenous secretion might be stimulated or inhibited differently in women in labor with OT infusion depending on the variation of basal OT levels.

In addition, oxytocinase, a placenta-derived enzyme, exerts a major effect on the clearance of OT in pregnant women and might increase the OT degradation during labor [27–29]. In laboring women with obesity, increased oxytocinase levels after OT infusion compared with pre-infusion levels have been demonstrated [12].

The method of choice for analyzing OT could also be of importance. Previous studies using different immunoassay methods for analyzing OT in pregnancy and labor, showed contradictory observations. Some demonstrated a dose dependent pattern while others did not [12,13]. The more specific MS/MS method has advantages over immunoassay methods, which are known to have significant variability [14,15] and vulnerability to interference from other peptides. Hence, a physiological reason for the non-consistent dose dependent pattern in maternal plasma OT levels, as demonstrated in the present study, might be more likely than deficiency in the measurement technique.

A major strength of this study is the novel design measuring serial plasma OT levels in women according to BMI before and during treatment with OT infusion in the first stage of active labor. Furthermore, the pre-analytical arrangement with a research team, including specially trained midwives and the authors working as obstetricians, working around the clock on the study, made it possible to maintain a highly standardized work procedure throughout the study. Moreover, the MS/MS method for analyzing OT levels, previously used as a validated assay for measuring human OT levels [30,31], was successfully introduced in this study for measuring maternal plasma OT levels during OT infusion in labor. The MS/MS method allows for absolute quantitation of the amount of OT molecules with high specificity and sensitivity, and with a low risk of interference and incorrect data.

There are also limitations to the study. In this pilot project a limited number of 40 women were recruited. Since the number of plasma samples in the analyses decreased with increasing OT infusion rate, the sample size was low at higher OT infusion rates. We only had access to a sufficiently large research team to include and follow patients with continuous sampling around the clock for a limited time period. Furthermore, even if MS/MS may be considered the method of choice when detecting and quantifying specific molecules, the method has limitations. Only the intact OT molecule is measured, the partly degraded or conjugated OT molecules will not be detected. There could also be individual samples that present with matrix effects that give ion suppression (this would probably be compensated for using the stable isotope labelled 'internal standard'). Another potential limitation is that aprotinin, a protease inhibitor, was not used during sample collection. However, in order to avoid extra additions (which could lead to an increased variability) we have instead decided to follow a very strict protocol for sampling, storing, sample handling and analysis including the use of a stable isotope labelled internal standard.

Since OT is released in pulses during labor [32], it is possible that some women had a peak in endogenous OT just before the plasma sample was taken, which might have influenced the OT pattern seen in the present study.

## Conclusions

The present study did not show any clear differences between women with non-obesity and obesity regarding plasma OT concentrations during labor and the OT levels varied

substantially both intra- and inter-individually in all BMI groups, with both rising and falling post infusion levels. Hence, these results do not support the hypothesis that women with obesity have low plasma OT levels during labor, suggesting there may be other factors than the plasma OT levels that determine the clinical response to OT infusion in women with obesity.

Further, no dose dependent increase in OT levels during OT infusion in labor was seen. It is possible that the observed clinical need and individual response in women with obesity would be a better predictor of different plasma OT levels rather than a pre-determined OT infusion rate. Guidelines for labor augmentation with OT may therefore be individualized according to the clinical response rather than generalized.

The reasons for the need for higher doses of OT infusion in women with obesity in labor remains unknown. To bring more clarity and improve the clinical care in the increasing group of women with obesity in maternal health care, future studies should be encouraged to focus on OT levels in relation to the clinical response rather than on the OT infusion rate.

## Supporting information

**S1 Table. Individual serial levels of plasma OT in the study population (n = 40) before and during OT infusion according to BMI.**
(DOCX)

## Acknowledgments

Our sincere gratitude goes to the women participating in the study. The midwives in the OLIV research team are acknowledged for their work on the clinical part of the study. Karin Söderman is acknowledged for her support with practical help and advice concerning the blood sampling.

## Author Contributions

**Conceptualization:** Anna Ramö Isgren, Sara Carlhäll, Preben Kjölhede, Jonas Bergquist, Marie Blomberg.

**Data curation:** Jonas Bergquist, Marie Blomberg.

**Formal analysis:** Marie Blomberg.

**Funding acquisition:** Jonas Bergquist, Marie Blomberg.

**Methodology:** Anna Ramö Isgren, Sara Carlhäll, Mark Dennis Retrato, Chamali Kodikara, Kumari A. Ubhayasekera, Preben Kjölhede, Jonas Bergquist, Marie Blomberg.

**Project administration:** Anna Ramö Isgren.

**Software:** Marie Blomberg.

**Supervision:** Jonas Bergquist, Marie Blomberg.

**Validation:** Mark Dennis Retrato, Kumari A. Ubhayasekera, Jonas Bergquist.

**Writing – original draft:** Anna Ramö Isgren, Sara Carlhäll.

**Writing – review & editing:** Mark Dennis Retrato, Chamali Kodikara, Kumari A. Ubhayasekera, Preben Kjölhede, Jonas Bergquist, Marie Blomberg.

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
