## [Decision Letter · Decision Letter 0]

7 Jul 2023

PONE-D-23-14377The association between maternal body mass index and serial plasma oxytocin levels during labor.PLOS ONE

Dear Dr. Blomberg,

Thank you for submitting your manuscript to PLOS ONE. After careful consideration, we feel that it has merit but does not fully meet PLOS ONE’s publication criteria as it currently stands. Therefore, we invite you to submit a revised version of the manuscript that addresses the points raised during the review process.

We look forward to receiving your revised manuscript.

Kind regards,

Alon Ben David

Academic Editor

PLOS ONE

Journal Requirements:

6. Please upload a new copy of Figure 4 and 5 as the detail is not clear. Please follow the link for more information: 

https://blogs.plos.org/plos/2019/06/looking-good-tips-for-creating-your-plos-figures-graphics/

https://blogs.plos.org/plos/2019/06/looking-good-tips-for-creating-your-plos-figures-graphics/

**Additional Editor Comments:**

This study aims to evaluate whether plasma levels of exogenous oxytocin, used for various indications as a uterotonic agent during labor, is affected by BMI. This is an important yet unanswered question. The article is interesting, well written and follows the journal's publishing policies although assessment could not be complete due to the low quality of some of the figures.

Please see reviewer comments below.

Reviewers' comments:

Reviewer's Responses to Questions

**Comments to the Author**

1. Is the manuscript technically sound, and do the data support the conclusions?

Reviewer #1: Yes

2. Has the statistical analysis been performed appropriately and rigorously? 

Reviewer #1: Yes

3. Have the authors made all data underlying the findings in their manuscript fully available?

Reviewer #1: No

4. Is the manuscript presented in an intelligible fashion and written in standard English?

Reviewer #1: Yes

5. Review Comments to the Author

Reviewer #1: I had the pleasure of reviewing the manuscript by Isgren et al. that reports baseline plasma OT concentration in laboring women stratified by weight, and their relative response to standardized OT infusion. The authors report low positive correlation between plasma OT and BMI at baseline, and increased plasma OT in high BMI subjects after OT infusion at 3.3 mU/min, but not at higher dose ranges. I commend the authors for this sample-rich study using a sophisticated technique that is more specific than commercially available OT ELISA assays. I would like the authors to address the following:

1. Aprotinin, a protease inhibitor, was not used during sample collection. It has been recommended to use aprotinin for these assays. The authors can cite this as a limitation.

2. None of the figures are high resolution, so I am unable to interpret any of the included figures. I would urge the authors to submit individual figure files for final review as this is the crux of their research.

3. The authors can include a few references for the use of tandem mass spectrometry as a valid assay for OT.

6. PLOS authors have the option to publish the peer review history of their article (what does this mean?). If published, this will include your full peer review and any attached files.

Reviewer #1: No

---

## [Author Response · Author response to Decision Letter 0]

19 Jul 2023

Response to Reviewers

Additional Editor Comments:

This study aims to evaluate whether plasma levels of exogenous oxytocin, used for various indications as a uterotonic agent during labor, is affected by BMI. This is an important yet unanswered question. The article is interesting, well written and follows the journal's publishing policies although assessment could not be complete due to the low quality of some of the figures.

Please see reviewer comments below.

Answer: Thank you

Reviewers' comments:

Reviewer's Responses to Questions

Comments to the Author

1. Is the manuscript technically sound, and do the data support the conclusions?

Reviewer #1: Yes

Answer: Thank you

2. Has the statistical analysis been performed appropriately and rigorously? 

Reviewer #1: Yes

Answer: Thank you

3. Have the authors made all data underlying the findings in their manuscript fully available?

Reviewer #1: No

Answer: The dataset's DOI written in the manuscript should be provided by Dryad (an open data publishing platform) to the journal if they will publish the DOI in a data availability statement in the final published article.

This is the temporary link to share the data files with the journal office and reviewers: https://datadryad.org/stash/share/47uc8ATlVzTvchC2L8FG5gcZy6QMsK0ekSPU5essYbY

4. Is the manuscript presented in an intelligible fashion and written in standard English?

Reviewer #1: Yes

Answer: Thank you

5. Review Comments to the Author

Reviewer #1: I had the pleasure of reviewing the manuscript by Isgren et al. that reports baseline plasma OT concentration in laboring women stratified by weight, and their relative response to standardized OT infusion. The authors report low positive correlation between plasma OT and BMI at baseline, and increased plasma OT in high BMI subjects after OT infusion at 3.3 mU/min, but not at higher dose ranges. I commend the authors for this sample-rich study using a sophisticated technique that is more specific than commercially available OT ELISA assays. I would like the authors to address the following:

1. Aprotinin, a protease inhibitor, was not used during sample collection. It has been recommended to use aprotinin for these assays. The authors can cite this as a limitation.

Answer: This has been added as a potential limitation in the text, but in order to avoid extra additions (which could lead to an increased variability) we have instead decided to follow a very strict protocol for sampling, storing, sample handling and analysis including the use of a stable isotope labelled internal standard. 

2. None of the figures are high resolution, so I am unable to interpret any of the included figures. I would urge the authors to submit individual figure files for final review as this is the crux of their research.

Answer: We apologize for the inappropriate quality of the figures. Individual figure files have been uploaded after using PACE to ensure that all figures meet PLOS requirements. 

3. The authors can include a few references for the use of tandem mass spectrometry as a valid assay for OT.

Answer: We agree, two references have now been added to the manuscript. Hering et al. Approaches to improve the quantitation of oxytocin in human serum by mass spectrometry. Front. Chem. 2022 and Franke et al. Oxytocin analysis from human serum, urine, and saliva by orbitrap LCMS. Drug Test Anal. 2019. ________________________________________

6. PLOS authors have the option to publish the peer review history of their article (what does this mean?). If published, this will include your full peer review and any attached files.

Do you want your identity to be public for this peer review? For information about this choice, including consent withdrawal, please see our Privacy Policy.

Reviewer #1: No

---

## [Editor Report · Decision Letter 1]

1 Aug 2023

The association between maternal body mass index and serial plasma oxytocin levels during labor.

PONE-D-23-14377R1

Dear Dr. Blomberg,

We’re pleased to inform you that your manuscript has been judged scientifically suitable for publication and will be formally accepted for publication once it meets all outstanding technical requirements.

Kind regards,

Alon Ben David

Academic Editor

PLOS ONE

---

## [Editor Report · Acceptance letter]

4 Aug 2023

PONE-D-23-14377R1 

The association between maternal body mass index and serial plasma oxytocin levels during labor. 

Dear Dr. Blomberg:

I'm pleased to inform you that your manuscript has been deemed suitable for publication in PLOS ONE. Congratulations! Your manuscript is now with our production department. 

Kind regards, 

on behalf of

Dr. Alon Ben David 

Academic Editor

PLOS ONE